# Market Scoring Rules Act As Opinion Pools For Risk-Averse Agents

**Mithun Chakraborty, Sanmay Das**
Department of Computer Science and Engineering
Washington University in St. Louis
St. Louis, MO 63130
{mithunchakraborty,sanmay}@wustl.edu

## Abstract

A market scoring rule (MSR) – a popular tool for designing algorithmic prediction markets – is an incentive-compatible mechanism for the aggregation of probabilistic beliefs from myopic risk-neutral agents. In this paper, we add to a growing body of research aimed at understanding the precise manner in which the price process induced by a MSR incorporates private information from agents who deviate from the assumption of risk-neutrality. We first establish that, for a myopic trading agent with a risk-averse utility function, a MSR satisfying mild regularity conditions elicits the agent's risk-neutral probability conditional on the latest market state rather than her true subjective probability. Hence, we show that a MSR under these conditions effectively behaves like a more traditional method of belief aggregation, namely an *opinion pool*, for agents' true probabilities. In particular, the logarithmic market scoring rule acts as a logarithmic pool for constant absolute risk aversion utility agents, and as a linear pool for an atypical budget-constrained agent utility with decreasing absolute risk aversion. We also point out the interpretation of a market maker under these conditions as a Bayesian learner even when agent beliefs are static.

## 1 Introduction

How should we combine opinions (or beliefs) about hidden truths (or uncertain future events) furnished by several individuals with potentially diverse information sets into a single group judgment for decision or policy-making purposes? This has been a fundamental question across disciplines for a long time (Surowiecki [2005]). One simple, principled approach towards achieving this end is the *opinion pool* (OP) which directly solicits inputs from informants in the form of probabilities (or distributions) and then maps this vector of inputs to a single probability (or distribution) based on certain axioms (Genest and Zidek [1986]). However, this technique abstracts away from the issue of providing proper incentives to a selfish-rational agent to reveal her private information honestly. Financial markets approach the problem differently, offering financial incentives for traders to supply their information about valuations and aggregating this information into informative prices. A *prediction market* is a relatively novel tool that builds upon this idea, offering trade in a financial security whose final monetary worth is tied to the future revelation of some currently unknown ground truth. Hanson [2003] introduced a family of algorithms for designing automated prediction markets called the *market scoring rule* (MSR) of which the Logarithmic Market Scoring Rule (LMSR) is arguably the most widely used and well-studied. A MSR effectively acts as a cost function-based market maker always willing to take the other side of a trade with any willing buyer or seller, and re-adjusting its quoted price after every transaction.

One of the most attractive properties of a MSR is its incentive-compatibility for a myopic risk-neutral trader. But this also means that, every time a MSR trades with such an agent, the updated market

price is reset to the subjective probability of that agent; the market mechanism itself does not play an active role in unifying pieces of information gleaned from the entire trading history into its current price. Ostrovsky [2012] and Iyer et al. [2014] have shown that, with differentially informed Bayesian risk-neutral and risk-averse agents respectively, trading repeatedly, "information gets aggregated" in a MSR-based market in a perfect Bayesian equilibrium. However, if agent beliefs themselves do not converge, can the price process emerging out of their interaction with a MSR still be viewed as an aggeragator of information in some sense? Intuitively, even if an agent does not revise her belief based on her inference about her peers' information from market history, her conservative attitude towards risk should compel her to trade in such a way as to move the market price not all the way to her private belief but to some function of her belief and the most recent price; thus, the evolving price should always retain some memory of all agents' information sequentially injected into the market. Therefore, the assumption of belief-updating agents may not be indispensable for providing theoretical guarantees on how the market incorporates agent beliefs. A few attempts in this vein can be found in the literature, typically embedded in a broader context (Sethi and Vaughan [2015], Abernethy et al. [2014]), but there have been few general results; see Section 1.1 for a review.

In this paper, we develop a new unified understanding of the information aggregation characteristics of a market with risk-averse agents mediated by a MSR, with no regard to how the agents' beliefs are formed. In fact, we demonstrate an equivalence between such MSR-mediated markets and opinion pools. We do so by first proving, in Section 3, that for any MSR interacting with myopic risk-averse traders, the revised instantaneous price after every trade equals the latest trader's risk-neutral probability conditional on the preceding market state. We then show that this price update rule satisfies an axiomatic characterization of opinion pooling functions from the literature, establishing the equivalence. In Sections 3.1, and 3.2, we focus on a specific MSR, the commonly used logarithmic variety (LMSR). We demonstrate that a LMSR-mediated market with agents having constant absolute risk aversion (CARA) utilities is equivalent to a logarithmic opinion pool, and that a LMSR-mediated market with budget-constrained agents having a specific concave utility with decreasing absolute risk aversion is equivalent to a linear opinion pool. We also demonstrate how the agents' utility function parameters acquire additional significance with respect to this pooling operation, and that in these two scenarios the market maker can be interpreted as a Bayesian learning algorithm *even if agents never update beliefs*. Our results are reminiscent of similar findings about competitive equilibrium prices in markets with rational, risk-averse agents (Pennock [1999], Beygelzimer et al. [2012], Millin et al. [2012] etc.), but those models require that agents learn from prices and also abstract away from any consideration of microstructure and the dynamics of actual price formation (how the agents would reach the equilibrium is left open). By contrast, our results do not presuppose any kind of generative model for agent signals, and also do not involve an equilibrium analysis – hence they can be used as tools to analyze the convergence characteristics of the market price in non-equilibrium situations with potentially fixed-belief or irrational agents.

## 1.1 Related Work

Given the plethora of experimental and empirical evidence that prediction markets are at least as effective as more traditional means of belief aggregation (Wolfers and Zitzewitz [2004], Cowgill and Zitzewitz [2013]), there has been considerable work on understanding how such a market formulates its own consensus belief from individual signals. An important line of research (Beygelzimer et al. [2012], Millin et al. [2012], Hu and Storkey [2014], Storkey et al. [2015]) has focused on a competitive equilibrium analysis of prediction markets under various trader models, and found an equivalence between the market's equilibrium price and the outcome of an opinion pool with the same agents. Seminal work in this field was done by Pennock [1999] who showed that linear and logarithmic opinion pools arise as special cases of the equilibrium of his intuitive model of securities markets when all agents have generalized logarithmic and negative exponential utilities respectively. Unlike these analyses that abstract away from the microstructure, Ostrovsky [2012] and Iyer et al. [2014] show that certain market structures (including market scoring rules) satisfying mild conditions perform "information aggregation" (i.e. the market's belief measure converges in probability to the ground truth) for repeatedly trading and learning agents with risk-neutral and risk-averse utilities respectively. Our contribution, while drawing inspiration from these sources, differs in that we delve into the characteristics of the evolution of the price rather than the properties of prices in equilibrium, and examine the manner in which the microstructure induces aggregation even if the agents are not Bayesian. While there has also been significant work on market properties in

continuous double auctions or markets mediated by sophisticated market-making algorithms (e.g. Cliff and Bruten [1997], Farmer et al. [2005], Brahma et al. [2012] and references therein) when the agents are "zero intelligence" or derivatives thereof (and therefore definitely not Bayesian), this line of literature has not looked at market scoring rules in detail, and analytical results have been rare.

In recent years, the literature focusing on the market scoring rule (or, equivalently, the cost function-based market maker) family has grown substantially. Chen and Vaughan [2010] and Frongillo et al. [2012] have uncovered isomorphisms between this type of market structure and well-known machine learning algorithms. We, on the other hand, are concerned with the similarities between price evolution in MSR-mediated markets and opinion pooling methods (see e.g. Garg et al. [2004]). Our work comes close to that of Sethi and Vaughan [2015] who show analytically that the price sequence of a cost function-based market maker with budget-limited risk-averse traders is "convergent under general conditions", and by simulation that the limiting price of LMSR with multi-shot but myopic logarithmic utility agents is approximately a linear opinion pool of agent beliefs. Abernethy et al. [2014] show that a risk-averse exponential utility agent with an exponential family belief distribution updates the state vector of a generalization of LMSR that they propose to a convex combination of the current market state vector and the natural parameter vector of the agent's own belief distribution (see their Theorem 5.2, Corollary 5.3) – this reduces to a logarithmic opinion pool (LogOP) for classical LMSR. The LMSR-LogOP connection was also noted by Pennock and Xia [2011] (in their Theorem 1) but with respect to an artificial probability distribution based on an agent's observed trade that the authors defined instead of considering traders' belief structure or strategies. We show how results of this type arise as special cases of a more general MSR-OP equivalence that we establish in this paper.

## 2 Model and definitions

Consider a decision-maker or *principal* interested in the "opinions" / "beliefs" / "forecasts" of a group of $n$ agents about an extraneous random binary event $X \in \{0, 1\}$, expressed in the form of point probabilities $\pi_i \in (0, 1)$, $i = 1, 2, \ldots, n$, i.e. $\pi_i$ is agent $i$'s subjective probability $\Pr(X = 1)$. $X$ can represent a proposition such as "A Republican will win the next U.S. presidential election" or "The favorite will beat the underdog by more than a pre-determined point spread in a game of football" or "The next *Avengers* movie will hit a certain box office target in its opening week." In this section, we briefly describe two approaches towards the aggregation of such private beliefs: (1) the opinion pool, which disregards the problem of incentivizing truthful reports, and focuses simply on unifying multiple probabilistic reports on a topic, and (2) the market scoring rule, an incentive-based mechanism for extracting honest beliefs from selfish-rational agents.

### 2.1 Opinion Pool (OP)

This family of methods takes as input the vector of probabilistic reports $p_i$, $i = 1, 2, \cdots, n$ submitted by $n$ agents, also called *experts* in this context, and computes an aggregate or consensus operator $\hat{p} = f(p_1, p_2, \cdots, p_n) \in [0, 1]$. Garg et al. [2004] identified three desiderata for an opinion pool (other criteria are also recognized in the literature, but the following are the most basic and natural):

1. **Unanimity:** If all experts agree, the aggregate also agrees with them.

2. **Boundedness:** The aggregate is bounded by the extremes of the inputs.

3. **Monotonicity:** If one expert changes her opinion in a particular direction while all other experts' opinions remain unaltered, then the aggregate changes in the same direction.

**Definition 1.** *We call $\hat{p} = f(p_1, p_2, \cdots, p_n)$ a* valid *opinion pool for $n$ probabilistic reports if it possesses properties 1, 2, and 3 listed above.*

It is easy to derive the following result for recursively defined pooling functions that will prove useful for establishing an equivalence between market scoring rules and opinion pools. The proof is in Section 1 of the Supplementary Material.

**Lemma 1.** *For a two-outcome scenario, if $f_2(r_1, r_2)$ and $f_{n-1}(q_1, q_2, \ldots, q_{n-1})$ are valid opinion pools for two probabilistic reports $r_1, r_2$ and $n-1$ probabilistic reports $q_1, q_2, \ldots, q_{n-1}$ respectively, then $f(p_1, p_2, \ldots, p_n) = f_2(f_{n-1}(p_1, p_2, \ldots, p_{n-1}), p_n)$ is also a valid opinion pool for $n$ reports.*

Two popular opinion pooling methods are the Linear Opinion Pool (LinOP) and the Logarithmic Opinion Pool (LogOP) which are essentially a weighted average (or convex combination) and a renormalized weighted geometric mean of the experts' probability reports respectively.

$$\text{LinOP}(p_1, p_2, \cdots, p_n) = \sum_{i=1}^n \omega_i^{\text{lin}} p_i,$$

$$\text{LogOP}(p_1, p_2, \cdots, p_n) = \prod_{i=1}^n p_i^{\omega_i^{\text{log}}} \bigg/ \left[ \prod_{i=1}^n p_i^{\omega_i^{\text{log}}} + \prod_{i=1}^n (1 - p_i)^{\omega_i^{\text{log}}} \right],$$

for a two-outcome scenario, where $\omega_i^{\text{lin}}, \omega_i^{\text{log}} \geq 0 \; \forall i = 1, 2, \ldots, n$, $\sum_{i=1}^n \omega_i^{\text{lin}} = 1$, $\sum_{i=1}^n \omega_i^{\text{log}} = 1$.

## 2.2 Market Scoring Rule (MSR)

In general, a scoring rule is a function of two variables $s(\mathbf{p}, x) \in \mathbb{R} \cup \{-\infty, \infty\}$, where $\mathbf{p}$ is an agent's probabilistic prediction (density or mass function) about an uncertain event, $x$ is the realized or revealed outcome of that event after the prediction has been made, and the resulting value of $s$ is the agent's *ex post* compensation for prediction. For a binary event $X$, a scoring rule can just be represented by the pair $(s_1(p), s_0(p))$ which is the vector of agent compensations for $\{X = 1\}$ and $\{X = 0\}$ respectively, $p \in [0, 1]$ being the agent's *reported* probability of $\{X = 1\}$ which may or may not be equal to her *true* subjective probability, say, $\pi = \Pr(X = 1)$. A scoring rule is defined to be *strictly proper* if it is incentive-compatible for a *risk-neutral* agent, i.e. an agent maximizes her subjective expectation of her ex post compensation by reporting her true subjective probability: $\pi = \arg\max_{p \in [0,1]} [\pi s_1(p) + (1 - \pi) s_0(p)], \forall \pi \in [0, 1]$.

In addition, a two-outcome scoring rule is *regular* if $s_j(\cdot)$ is real-valued except possibly that $s_0(1)$ or $s_1(0)$ is $-\infty$; any regular strictly proper scoring rule can written in the following form (Gneiting and Raftery [2007]):

$$s_j(p) = G(p) + G'(p)(j - p), \quad j \in \{0, 1\}, p \in [0, 1], \tag{1}$$

$G : [0, 1] \to \mathbb{R}$ is a strictly convex function with $G'(\cdot)$ as a sub-gradient which is real-valued expect possibly that $-G'(0)$ or $G'(1)$ is $\infty$; if $G(\cdot)$ is differentiable in $(0, 1)$, $G'(\cdot)$ is simply its derivative.

A classic example of a regular strictly proper scoring rule is the logarithmic scoring rule:

$$s_1(p) = b \ln p; \quad s_0(p) = b \ln(1 - p), \quad \text{where } b > 0 \text{ is a free parameter.} \tag{2}$$

Hanson [2003] introduced an extension of a scoring rule wherein the principal initiates the process of information elicitation by making a baseline report $p_0$, and then elicits publicly declared reports $p_i$ sequentially from $n$ agents; the ex post compensation $c_x(p_i, p_{i-1})$ received by agent $i$ from the principal, where $x$ is the realized outcome of event $X$, is the difference between the scores assigned to the reports made by herself and her predecessor:

$$c_x(p_i, p_{i-1}) \triangleq s_x(p_i) - s_x(p_{i-1}), \quad x \in \{0, 1\}. \tag{3}$$

If each agent acts non-collusively, risk-neutrally, and myopically (as if her current interaction with the principal is her last), then the incentive compatibility property of a strictly proper score still holds for the sequential version. Moreover, it is easy to show that the principal's worst-case payout (loss) is bounded regardless of agent behavior. In particular, for the binary-outcome logarithmic score, the loss bound for $p_0 = 1/2$ is $b \ln 2$; $b$ can be referred to as the principal's *loss parameter*.

A sequentially shared strictly proper scoring rule of the above form can also be interpreted as a cost function-based prediction market mechanism offering trade in an Arrow-Debreu (i.e. $(0, 1)$-valued) security written on the event $X$, hence the name "market scoring rule". The cost function is a strictly convex function of the total outstanding quantity of the security that determines all execution costs; its first derivative (the cost per share of buying or the proceeds per share from selling an infinitesimal quantity of the security) is called the market's "instantaneous price", and can be interpreted as the market maker's current *risk-neutral probability* (Chen and Pennock [2007]) for $\{X = 1\}$, the starting price being equal to the principal's baseline report $p_0$. Trading occurs in discrete episodes $1, 2, \ldots, n$, in each of which an agent orders a quantity of the security to buy or sell given the market's cost function and the (publicly displayed) instantaneous price. Since there is a one-to-one correspondence between agent $i$'s order size and $p_i$, the market's revised instantaneous price after trading with agent $i$, an agent's "action" or trading decision in this setting is identical to making a probability report by selecting a $p_i \in [0, 1]$. If agent $i$ is risk-neutral, then $p_i$ is, by design, her subjective probability $\pi_i$ (see Hanson [2003], Chen and Pennock [2007] for further details).

**Definition 2.** *We call a market scoring rule* well-behaved *if the underlying scoring rule is regular and strictly proper, and the associated convex function $G(\cdot)$ (as in (1)) is continuous and thrice-differentiable, with $0 < G''(p) < \infty$ and $|G'''(p)| < \infty$ for $0 < p < 1$.*

## 3 MSR behavior with risk-averse myopic agents

We first present general results on the connection between sequential trading in a MSR-mediated market with risk-averse agents and opinion pooling, and then give a more detailed picture for two representative utility functions without and with budget constraints respectively. Please refer to Section 2 of the Supplementary Material for detailed proofs of all results in this section.

Suppose that, in addition to a belief $\pi_i = \Pr(X = 1)$, each agent $i$ has a continuous utility function of wealth $u_i(c)$, where $c \in [c_i^{\min}, \infty]$ denotes her (ex post) wealth, i.e. her net compensation from the market mechanism after the realization of $X$ defined in (3), and $c_i^{\min} \in [-\infty, 0]$ is her minimum acceptable wealth (a negative value suggests tolerance of debt); $u_i(\cdot)$ satisfies the usual criteria of *non-satiation* i.e. $u_i'(c) > 0$ except possibly that $u_i'(\infty) = 0$, and *risk aversion*, i.e. $u_i''(c) < 0$ except possibly that $u_i''(\infty) = 0$, through out its domain (Mas-Colell et al. [1995]); in other words $u_i(\cdot)$ is strictly increasing and strictly concave. Additionally, we require its first two derivatives to be finite and continuous on $[c_i^{\min}, \infty]$ except that we tolerate $u_i'(c_i^{\min}) = \infty$, $u_i''(c_i^{\min}) = -\infty$. Note that, by choosing a finite lower bound $c_i^{\min}$ on the agent's wealth, we can account for any starting wealth or budget constraint that effectively restricts the agent's action space.

**Lemma 2.** *If $|c_i^{\min}| < \infty$, then there exist lower and upper bounds, $p_i^{\min} \in [0, p_{i-1}]$ and $p_i^{\max} \in [p_{i-1}, 1]$ respectively, on the feasible values of the price $p_i$ to which agent $i$ can drive the market regardless of her belief $\pi_i$, where $p_i^{\min} = s_1^{-1}(c_i^{\min} + s_1(p_{i-1}))$ and $p_i^{\max} = s_0^{-1}(c_i^{\min} + s_0(p_{i-1}))$.*

Since the latest price $p_{i-1}$ can be viewed as the market's current "state" from myopic agent $i$'s perspective, the agent's final utility depends not only on her own action $p_i$ and the extraneously determined outcome $x$ but also on the current market state $p_{i-1}$ she encounters, her rational action being given by $p_i = \arg\max_{p \in [0,1]} [\pi_i u_i(c_1(p, p_{i-1})) + (1 - \pi_i)u_i(c_0(p_i, p_{i-1}))]$. This leads us to the main result of this section.

**Theorem 1.** *If a well-behaved market scoring rule for an Arrow-Debreu security with a starting instantaneous price $p_0 \in (0, 1)$ trades with a sequence of $n$ myopic agents with subjective probabilities $\pi_1, \ldots, \pi_n \in (0, 1)$ and risk-averse utility functions of wealth $u_1(\cdot), \ldots, u_n(\cdot)$ as above, then the updated market price $p_i$ after every trading episode $i \in \{1, 2, \ldots, n\}$ is equivalent to a valid opinion pool for the market's initial baseline report $p_0$ and the subjective probabilities $\pi_1, \pi_2, \ldots, \pi_i$ of all agents who have traded up to (and including) that episode.*

**Proof sketch.** For every trading epsiode $i$, by setting the first derivative of agent $i$'s expected utility to zero, and analyzing the resulting equation, we can arrive at the following lemmas.

**Lemma 3.** *Under the conditions of Theorem 1, if $p_{i-1} \in (0, 1)$, then the revised price $p_i$ after agent $i$ trades is the unique solution in $(0, 1)$ to the fixed-point equation:*

$$p_i = \frac{\pi_i u_i'(c_1(p_i, p_{i-1}))}{\pi_i u_i'(c_1(p_i, p_{i-1})) + (1 - \pi_i)u_i'(c_0(p_i, p_{i-1}))}. \tag{4}$$

Since $p_0 \in (0, 1)$, and $\pi_i \in (0, 1) \ \forall i$, $p_i$ is also confined to $(0, 1) \ \forall i$, by induction.

**Lemma 4.** *The implicit function $p_i(p_{i-1}, \pi_i)$ described by (4) has the following properties:*

1. *$p_i = \pi_i$ (or $p_{i-1}$) if and only if $\pi_i = p_{i-1}$.*

2. *$0 < \min\{p_{i-1}, \pi_i\} < p_i < \max\{p_{i-1}, \pi_i\} < 1$ whenever $\pi_i \neq p_{i-1}$, $0 < \pi_i, p_{i-1} < 1$.*

3. *For any given $p_{i-1}$ (resp. $\pi_i$), $p_i$ is a strictly increasing function of $\pi_i$ (resp. $p_{i-1}$).*

Evidently, properties 1, 2, and 3 above correspond to axioms of unanimity, boundedness, and monotonicity respectively, defined in Section 2. Hence, $p_i(p_{i-1}, \pi_i)$ is a valid opinion pooling function for $p_{i-1}, \pi_i$. Finally, since (4) defines the opinion pool $p_i$ recursively in terms of $p_{i-1} \ \forall i = 1, 2, \ldots, n$, we can invoke Lemma 1 to obtain the desired result. □

There are several points worth noting about this result. First, since the updated market price $p_i$ is also equivalent to agent $i$'s action (Section 2.2), the R.H.S. of (4) is agent $i$'s risk-neutral probability (Pennock [1999]) of $\{X = 1\}$, given her utility function, her action, and the current market state. Thus, Lemma 3 is a natural extension of the elicitation properties of a MSR. MSRs, by design, elicit subjective probabilities from risk-neutral agents in an incentive compatible manner; we show that, in general, they elicit risk-neutral probabilities when they interact with risk-averse agents. Lemma 3 is also consistent with the observation of Pennock [1999] that, for all belief elicitation schemes based on monetary incentives, an external observer can only assess a participant's risk-neutral probability uniquely; she cannot discern the participant's belief and utility separately. Second, observe that this pooling operation is accomplished by a MSR even without direct revelation. Finally, notice the presence of the market maker's own initial baseline $p_0$ as a component in the final aggregate; however, for the examples we study below, the impact of $p_0$ diminishes with the participation of more and more informed agents, and we conjecture that this is a generic property.

In general, the exact form of this pooling function is determined by the complex interaction between the MSR and agent utility, and a closed form of $p_i$ from (4) might not be attainable in many cases. However, given a paticular MSR, we can venture to identify agent utility functions which give rise to well-known opinion pools. Hence, for the rest of this paper, we focus on the logarithmic market scoring rule (LMSR), one of the most popular tools for implementing real-world prediction markets.

### 3.1 LMSR as LogOP for constant absolute risk aversion (CARA) utility

**Theorem 2.** *If myopic agent $i$, having a subjective belief $\pi_i \in (0, 1)$ and a risk-averse utility function satisfying our criteria, trades with a LMSR market with parameter $b$ and current instantaneous price $p_{i-1}$, then the market's updated price $p_i$ is identical to a logarithmic opinion pool between the current price and the agent's subjective belief, i.e.*

$$p_i = \pi_i^{\alpha_i} p_{i-1}^{1-\alpha_i} \big/ \left[ \pi_i^{\alpha_i} p_{i-1}^{1-\alpha_i} + (1-\pi_i)^{\alpha_i}(1-p_{i-1})^{1-\alpha_i} \right], \quad \alpha_i \in (0, 1), \tag{5}$$

*if and only if agent $i$'s utility function is of the form*

$$u_i(c) = \tau_i \left( 1 - \exp\left(-c/\tau_i\right) \right), \quad c \in \mathbb{R} \cup \{-\infty, \infty\}, \quad \text{constant } \tau_i \in (0, \infty), \tag{6}$$

*the aggregation weight being given by $\alpha_i = \frac{\tau_i/b}{1+\tau_i/b}$.*

The proof is in Section 2.1 of the Supplementary Material. Note that (6) is a standard formulation of the CARA (or negative exponential) utility function with *risk tolerance* $\tau_i$; smaller the value of $\tau_i$, higher is agent $i$'s aversion to risk. The unbounded domain of $u_i(\cdot)$ indicates a lack of budget constraints; risk aversion comes about from the fact that the range of the function is bounded above (by its risk tolerance $\tau_i$) but not bounded below.

Moreover, the LogOP equation (5) can alternatively be expressed as a linear update in terms of *log-odds ratios*, another popular means of formulating one's belief about a binary event:

$$l(p_i) = \alpha_i l(\pi_i) + (1-\alpha_i) l(p_{i-1}), \quad l(p) = \ln\left(\frac{p}{1-p}\right) \in [-\infty, \infty] \quad \text{for } p \in [0, 1]. \tag{7}$$

**Aggregation weight and risk tolerance:** Since $\alpha_i$ is an increasing function of an agent's risk tolerance relative to the market's loss parameter (the latter being, in a way, a measure of how much risk the market maker is willing to take), identity (7) implies that the higher an agent's risk tolerance, the larger is the contribution of her belief towards the changed market price, which agrees with intuition. Also note the interesting manner in which the market's loss parameter effectively scales down an agent's risk tolerance, enhancing the inertia factor $(1 - \alpha_i)$ of the price process.

**Bayesian interpretation:** The Bayesian interpretation of LogOP in general is well-known (Bordley [1982]); we restate it here in a form that is more appropriate for our prediction market setting. We can recast (5) as $p_i = p_{i-1} \left(\frac{\pi_i}{p_{i-1}}\right)^{\alpha_i} \big/ \left[ p_{i-1} \left(\frac{\pi_i}{p_{i-1}}\right)^{\alpha_i} + (1-p_{i-1}) \left(\frac{1-\pi_i}{1-p_{i-1}}\right)^{\alpha_i} \right]$. This shows that, over the $i^{\text{th}}$ trading episode $\forall i$, the LMSR-CARA agent market environment is equivalent to a Bayesian learner performing inference on the point estimate of the probability of the forecast event $X$, starting with the common-knowledge prior $\Pr(X = 1) = p_{i-1}$, and having direct access to $\pi_i$ (which corresponds to the "observation" for the inference problem), the likelihood function associated with this observation being $\mathcal{L}(X = x | \pi_i) \propto \left| \frac{1-x-\pi_i}{1-x-p_{i-1}} \right|^{\alpha_i}, x \in \{0, 1\}$.

**Sequence of one-shot traders:** If all $n$ agents in the system have CARA utilities with potentially different risk tolerances, and trade with LMSR myopically only once each in the order $1, \ldots, n$, then the "final" market log-odds ratio after these $n$ trades, on unfolding the recursion in (7), is given by $l(p_n) = \widetilde{\alpha}_0^n l(p_0) + \sum_{i=1}^n \widetilde{\alpha}_i^n l(\pi_i)$. This is a LogOP where $\widetilde{\alpha}_0^n = \prod_{i=1}^n (1 - \alpha_i)$ determines the inertia of the market's initial price, which diminishes as more and more traders interact with the market, and $\widetilde{\alpha}_j^n$, $j \geq 1$ quantifies the degree to which an individual trader impacts the final (aggregate) market belief; $\widetilde{\alpha}_j^n = \alpha_j \prod_{i=j+1}^n (1 - \alpha_i)$, $j = 1, \ldots, n-1$, and $\widetilde{\alpha}_n^n = \alpha_n$. Interestingly, the weight of an agent's belief depends not only on her own risk tolerance but also on those of all agents succeeding her in the trading sequence (lower weight for a more risk tolerant successor, ceteris paribus), and is independent of her predecessors' utility parameters. This is sensible since, by the design of a MSR, trader $i$'s belief-dependent action influences the action of each of (rational) traders $i+1, i+2, \ldots$ so that the action of each of these successors, in turn, has a role to play in determining the market impact of trader $i$'s belief. In particular, if $\tau_j = \tau > 0 \ \forall j \geq 1$, then the aggregation weights satisfy the inequalities $\widetilde{\alpha}_{j+1}^n / \widetilde{\alpha}_j^n = 1 + \tau/b > 1 \ \forall j = 1, \cdots, n-1$, i.e. LMSR assigns progessively higher weights to traders arriving later in the market's lifetime when they all exhibit identical constant risk aversion. This seems to be a reasonable aggregation principle in most scenarios wherein the amount of information in the world improves over time. Moreover, in this situation, $\widetilde{\alpha}_1^n / \widetilde{\alpha}_0^n = \tau/b$ which indicates that the weight of the market's baseline belief in the aggregate may be higher than those of some of the trading agents if the market maker has a comparatively high loss parameter. This strong effect of the trading sequence on the weights of agents' beliefs is a significant difference between the one-shot trader setting and the market equilibrium setting where each agent's weight is independent of the utility function parameters of her peers.

**Convergence:** If agents' beliefs are themselves independent samples from the same distribution $\mathcal{P}$ over $[0,1]$, i.e. $\pi_i \sim_{\text{i.i.d.}} \mathcal{P} \ \forall i$, then by the sum laws of expectation and variance,

$$\mathbb{E}\left[l(p_n)\right] = \widetilde{\alpha}_0^n l(p_0) + (1 - \widetilde{\alpha}_0^n)\mathbb{E}_{\pi \sim \mathcal{P}}\left[l(\pi)\right]; \quad \text{Var}\left[l(p_n)\right] = \text{Var}_{\pi \sim \mathcal{P}}\left[l(\pi)\right] \sum_{i=1}^n (\widetilde{\alpha}_i^n)^2.$$

Hence, using an appropriate concentration inequality (Boucheron et al. [2004]) and the properties of the $\widetilde{\alpha}_i^n$'s, we can show that, as $n$ increases, the market log-odds ratio $l(p_n)$ converges to $\mathbb{E}_{\pi \sim \mathcal{P}}\left[l(\pi)\right]$ with a high probability; this convergence guarantee does not require the agents to be Bayesian.

### 3.2 LMSR as LinOP for an atypical utility with decreasing absolute risk aversion

**Theorem 3.** *If myopic agent $i$, having a subjective belief $\pi_i \in (0,1)$ and a risk-averse utility function satisfying our criteria, trades with a LMSR market with parameter $b$ and current instantaneous price $p_{i-1}$, then the market's updated price $p_i$ is identical to a linear opinion pool between the current price and the agent's subjective belief, i.e.*

$$p_i = \beta_i \pi_i + (1 - \beta_i)p_{i-1}, \quad \text{for some constant } \beta_i \in (0,1), \tag{8}$$

*if and only if agent $i$'s utility function is of the form*

$$u_i(c) = \ln(\exp((c + B_i)/b) - 1), \quad c \geq -B_i, \tag{9}$$

*where $B_i > 0$ represents agent $i$'s budget, the aggregation weight being $\beta_i = 1 - \exp(-B_i/b)$.*

The proof is in Section 2.2 of the Supplementary Material. The above atypical utility function has its domain bounded below, and possesses a positive but strictly decreasing Arrow-Pratt absolute risk aversion measure (Mas-Colell et al. [1995]) $A_i(c) = -u_i''(c)/u_i'(c) = \frac{1}{b(\exp((c+B_i)/b)-1)}$ for any $b, B_i > 0$. It shares these characteristics with the well-known logarithmic utility function. Moreover, although this function is approximately linear for large (positive) values of the wealth $c$, it is approximately logarithmic when $(c + B_i) \ll b$.

Theorem 3 is somewhat surprising since it is logarithmic utility that has traditionally been found to effect a LinOP in a market equilibrium (Pennock [1999], Beygelzimer et al. [2012], Storkey et al. [2015], etc.). Of course in this paper, we are not in an equilibrium / convergence setting, but in light of the above similarities between utility function (9) and logarithmic utility, it is perhaps not unreasonable to ask whether the logarithmic utility-LinOP connection is still maintained approximately for LMSR price evolution under some conditions. We have extensively explored this idea, both analytically and by simulations, and have found that a small agent budget compared to the LMSR loss parameter $b$ seems to produce the desired result (see Section 3 of the Supplementary Material).

Note that, unlike in Theorem 2, the equivalence here requires the agent utility function to depend on the market maker's loss parameter $b$ (the scaling factor in the exponential). Since the microstructure is assumed to be common knowledge, as in traditional MSR settings, the consideration of an agent utility that takes into account the market's pricing function is not unreasonable.

Since the domain of utility function (9) is bounded below, we can derive $\pi_i$-independent bounds on possible values of $p_i$ from Lemma 2: $p_i^{\min} = (1 - \beta_i)p_{i-1}$, $p_i^{\max} = \beta_i + (1 - \beta_i)p_{i-1}$. Hence, equation (8) becomes $p_i = \pi_i p_i^{\max} + (1 - \pi_i)p_i^{\min}$, i.e. the revised price is a linear interpolation between the agent's price bounds, her subjective probability itself acting as the interpolation factor.

**Aggregation weight and budget constraint:** Evidently, the aggregation weight of agent $i$'s belief, $\beta_i = (1 - \exp(-B_i/b))$, is an increasing function of her budget normalized with respect to the market's loss parameter; it is, in a way, a measure of her relative risk tolerance. Thus, broad characteristics analogous to the ones in Section 3.1 apply to these aggregation weights as well, with the log-odds ratio replaced by the actual market price.

**Bayesian interpretation:** Under the mild technical assumption that agent $i$'s belief $\pi_i \in (0,1)$ is rational, and her budget $B_i > 0$ is such that $\beta_i \in (0,1)$ is also rational, it is possible to obtain positive integers $r_i$, $N_i$ and a positive rational number $m_{i-1}$ such that $\pi_i = r_i/N_i$ and $\beta_i = N_i/(m_{i-1}+N_i)$. Then, we can rewrite the LinOP equation (8) as $p_i = \frac{r_i + p_{i-1}m_{i-1}}{m_{i-1} + N_i}$, which is equivalent to the posterior expectation of a beta-binomial Bayesian inference procedure described as follows: The forecast event $X$ is modeled as the (future) final flip of a biased coin with an unknown probability of heads. In episode $i$, the principal (or aggregator) has a prior distribution $\text{BETA}(\mu_{i-1}, \nu_{i-1})$ over this probability, with $\mu_{i-1} = p_{i-1}m_{i-1}$, $\nu_{i-1} = (1 - p_{i-1})m_{i-1}$. Thus, $p_{i-1}$ is the prior mean and $m_{i-1}$ the corresponding "pseudo-sample size" parameter. Agent $i$ is non-Bayesian, and her subjective probability $\pi_i$, accessible to the aggregator, is her maximum likelihood estimate associated with the (binomial) likelihood of observing $r_i$ heads out of a private sample of $N_i$ independent flips of the above coin ($N_i$ is common knowledge). Note that $m_{i-1}$, $N_i$ are measures of certainty of the aggregator and the trading agent respectively, and the latter's normalized budget $B_i/b = \ln(1 + N_i/m_{i-1})$ becomes a measure of her certainty relative to the aggregator's current state in this interpretation.

**Sequence of one-shot traders and convergence:** If all agents have utility (9) with potentially different budgets, and trade with LMSR myopically once each, then the final aggregate market price is given by $p_n = \widetilde{\beta}_0^n p_0 + \sum_{i=1}^n \widetilde{\beta}_i^n \pi_i$, which is a LinOP where $\widetilde{\beta}_0^n = \prod_{i=1}^n (1 - \alpha_i)$, $\widetilde{\beta}_j^n = \beta_j \prod_{i=j+1}^n (1 - \beta_i) \ \forall j = 1, \ldots, n-1$, $\widetilde{\beta}_n^n = \beta_n$. Again, all intuitions about $\widetilde{\alpha}_j^n$ from Section 3.1 carry over to $\widetilde{\beta}_j^n$. Moreover, if $\pi_i \sim_{\text{i.i.d.}} \mathcal{P} \ \forall i$, then we can proceed exactly as in Section 3.1 to show that, as $n$ increases, $p_n$ converges to $\mathbb{E}_{\pi \sim \mathcal{P}}[\pi]$ with a high probability.

# 4  Discussion and future work

We have established the correspondence of a well-known securities market microstructure to a class of traditional belief aggregation methods and, by extension, Bayesian inference procedures in two important cases. An obvious next step is the identification of general conditions under which a MSR and agent utility combination is equivalent to a given pooling operation. Another research direction is extending our results to a sequence of agents who trade repeatedly until "convergence", taking into account issues such as the order in which agents trade when they return, the effects of the updated wealth after the first trade for agents with budgets, etc.

**Acknowledgments**

We are grateful for support from NSF IIS awards 1414452 and 1527037.

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
