[Supplementary Material]

# Supplementary Material: Market Scoring Rules Act As Opinion Pools For Risk-Averse Agents

**Mithun Chakraborty, Sanmay Das**
Department of Computer Science and Engineering
Washington University in St. Louis
St. Louis, MO 63130
{mithunchakraborty,sanmay}@wustl.edu

## Abstract

Here, we present detailed proofs of all theorems in the main paper, as well as some observations and experiments which we had to exclude from the main paper owing to paucity of space.

## 1 Model and definitions

Here, we provide the proof of Lemma 1 from Section 2 of the main paper.

**Restatement of Lemma 1.** *For a two-ouctome forecasting problem where an expert's report can be specified in terms of a single probability $p \in [0,1]$, if $f_2(r_1, r_2)$ and $f_{n-1}(q_1, q_2, \ldots, q_{n-1})$ are valid opinion pools for two probabilistic reports $r_1, r_2$ and $n-1$ probabilistic reports $q_1, q_2, \ldots, q_{n-1}$ respectively, then $f(p_1, p_2, \ldots, p_n) = f_2(f_{n-1}(p_1, p_2, \ldots, p_{n-1}), p_n)$ is also a valid opinion pool for $n$ reports.*

*Proof.* Recall from Definition 1 in the main paper that a valid opinion pool $\widehat{p} = \phi(p_1, p_2, \ldots, p_m)$, where $p_1, p_2, \ldots, p_m \in [0,1]$ are reported expert probabilities of occurrence of binary event $X$, must satisfy

1. **Unanimity:** If $p_i = p \ \forall i = 1, 2, \ldots, m$, then $\widehat{p} = p$.

2. **Boundedness:** $\min\{p_1, p_2, \ldots, p_m\} \leq \widehat{p} \leq \max\{p_1, p_2, \ldots, p_m\}$.

3. **Monotonicity:** $\widehat{p}$ increases monotonically as $p_i$ increases, $p_j$ being held constant $\forall j \neq i$, $i = 1, 2, \ldots, m$, i.e. $\frac{\partial \phi}{\partial p_i} > 0$ everywhere $\forall i$.

By the condition of the lemma, all the above three properties are possessed by each each of $f_2$ and $f_{n-1}$, and we need to prove that $f$ has each of these properties, too.

**To prove the unanimity of $f$:** Let $p_i = p \ \forall i = 1, 2, \ldots, n$. Then,

$$
\begin{aligned}
f(p, p, \ldots, p) &= f_2(f_{n-1}(p, p, \ldots, p), p) \\
&= f_2(p, p), \quad \text{by unanimity of } f_{n-1}, \\
&= p, \quad \text{by unanimity of } f_2.
\end{aligned}
$$

**To prove the boundedness of $f$:** Using the upper bounds on $f_2$ and $f_{n-1}$,

$$
\begin{aligned}
f(p_1, p_2, \ldots, p_n) &\leq \max\{f_{n-1}(p_1, p_2, \ldots, p_{n-1}), p_n\} \\
&\leq \max\{\max\{p_1, p_2, \ldots, p_{n-1}\}, p_n\} \\
&= \max\{p_1, p_2, \ldots, p_{n-1}, p_n\}.
\end{aligned}
$$

Similarly, using the lower bounds on $f_2$ and $f_{n-1}$, we can show that $f(p_1, p_2, \ldots, p_n) \geq \min\{p_1, p_2, \ldots, p_{n-1}, p_n\}$.

**To prove the monotonicity of $f$:** The partial derivative of $f$ with respect to each $p_i$, $i = 1, 2, \ldots, n-1$ is given by

$$
\frac{\partial f}{\partial p_i} = \frac{\partial}{\partial p_i} f_2(f_{n-1}(p_1, p_2, \ldots, p_{n-1}), p_n) = \frac{\partial f_2(f_{n-1}, p_n)}{\partial f_{n-1}} \cdot \frac{\partial f_{n-1}(p_1, p_2, \ldots, p_{n-1})}{\partial p_n} > 0
$$

by the monotonicity of $f_2$ and $f_{n-1}$ with respect to their respective inputs. Similarly,

$$
\frac{\partial f}{\partial p_n} = \frac{\partial f_2(f_{n-1}, p_n)}{\partial p_n} > 0
$$

by the monotonicity of $f_2$. $\qquad\square$

## 2 A general well-behaved MSR as an Opinion Pool for a general risk-averse utility

First, we shall recapitulate the mathematical properties of a *well-behaved* market scoring rule (Definition 2 in the main paper): The underlying (strictly proper and regular) scoring rule for such a MSR can be written as

$$
s_j(p) = \begin{cases} G(p) + G'(p)(j - p), & j \in \{0, 1\}, p \in [0, 1], p \neq j, \\ G(p), & p = j \in \{0, 1\} \end{cases} \tag{1}
$$

from (1) in the main paper, where

1. $G : [0, 1] \to \mathbb{R}$ is a continuous function.
2. $G'(\cdot)$ is real-valued in $[0, 1]$ except possibly that $G'(0) = -\infty$ or $G'(1) = \infty$.
3. $G''(\cdot)$ exists and is positive in $[0, 1]$, $0 < G''(p) < \infty$ for $0 < p < 1$.
4. $G'''(\cdot)$ exists, and $|G'''(p)| < \infty$ for $0 < p < 1$.

Notice that the positivity of $G''(\cdot)$ implies the strict convexity of $G(\cdot)$ and the increasing monotonicity of $G'(\cdot)$. Property 2 ensures that $s_j(\cdot)$ is real-valued except possibly that $s_0(1) = \infty$ or $s_1(0) = \infty$. $G(p) = ps_1(p) + (1 - p)s_0(1 - p)$ is the expected score function sometimes called the *information measure* or *generalized entropy function* associated with the scoring rule $s_x(\cdot)$ (Gneiting and Raftery [2007]).

For $x \in \{0, 1\}$, the first derivative of $s_x(p)$, $\forall p \in (0, 1)$, is

$$
s_x'(p) = G''(p)(x - p) \implies s_1'(p) = G''(p)(1 - p) > 0, \quad s_0'(p) = -G''(p)p < 0,
$$

since $G''(p) > 0$. Hence, $s_1(p)$ and $s_0(p)$ are strictly increasing and decreasing functions of $p$ respectively, which is quite intuitive since the reward for predicting a higher probability for the outcome that actually materialized should be higher.

Moreover, if $p_{i-1}$ and $p_i$ denote respectively the instantaneous price of a MSR immediately before and after agent $i$ interacts with it, then by the design of a MSR, the agent's ex post compensation from the market for any outcome $x \in \{0, 1\}$ is given by

$$
c_x(p_i, p_{i-1}) = s_x(p_i) - s_x(p_{i-1}).
$$

We can readily obtain the following properties of $c_x$:

$$
c_1(p, p_{i-1}) - c_0(p, p_{i-1}) = G'(p) - G'(p_{i-1}); \tag{2}
$$

$$
\frac{\partial}{\partial p} c_x(p, p_{i-1}) = s_x'(p) = G''(p)(x - p), \quad \forall p_{i-1} \in (0, 1), x \in \{0, 1\}. \tag{3}
$$

Hence, $c_1(p, p_{i-1})$ and $c_0(p, p_{i-1})$ are also strictly increasing and decreasing in $p$ respectively, regardless of $p_{i-1}$, as expected.

Next, we shall enumerate, from Section 3 in the main paper, the criteria that an agent utility function $u_i(\cdot)$ must meet in our setting:

1. **Continuity:** $u_i(\cdot)$ is continuous over $[c_i^{\min}, \infty]$ where $c_i^{\min}$ can attain any value in $[-\infty, 0]$.
2. **Increasing monotonicity (Non-satiation):** $u_i'(\cdot)$ is continuous and positive real-valued over $[c_i^{\min}, \infty]$ except possibly that $u_i'(c_{\min}) = \infty$ or $u_i'(\infty) = 0$.
3. **Strict concavity (Risk aversion):** $u_i''(\cdot)$ is continuous and negative real-valued over $[c_i^{\min}, \infty]$ except possibly that $u_i''(c_{\min}) = -\infty$ or $u_i''(\infty) = 0$.

The following is the proof of Lemma 2 from Section 3 of the main paper.

**Restatement of Lemma 2** *If $|c_i^{\min}| < \infty$, then there exist upper and lower bounds, $p_i^{\min} \in [0, p_{i-1}]$ and $p_i^{\max} \in [p_{i-1}, 1]$ respectively, on the feasible values of the price $p_i$ to which agent $i$ can drive the market from $p_{i-1} \in (0, 1)$ regardless of her belief $\pi_i$, where $p_i^{\min} = s_1^{-1}(c_i^{\min} + s_1(p_{i-1}))$ and $p_i^{\max} = s_0^{-1}(c_i^{\min} + s_0(p_{i-1}))$.*

*Proof.* Agent $i$'s ex post wealth for trading in such a way as to revise the market price from $p_{i-1}$ to any $\tilde{p} \in [0, 1]$ is $c_x(\tilde{p}, p_{i-1})$ for outcome $x$ but, from the constraints imposed by the utility function, this wealth cannot be smaller than $c_i^{\min}$ for any $x$. Thus,

$$c_1(\tilde{p}, p_{i-1}) \geq c_i^{\min}$$
$$\Rightarrow \quad s_1(\tilde{p}) - s_1(p_{i-1}) \geq c_i^{\min}$$
$$\Rightarrow \quad s_1(\tilde{p}) \geq c_i^{\min} + s_1(p_{i-1})$$
$$\Rightarrow \quad \tilde{p} \geq s_1^{-1}(c_i^{\min} + s_1(p_{i-1})) = p_i^{\min},$$

since $s_1(\cdot)$ is strictly increasing (hence invertible). Also, since $c_i^{\min} \leq 0$,

$$s_1(p_i^{\min}) = c_i^{\min} + s_1(p_{i-1}) \quad \implies \quad s_1(p_i^{\min}) \leq s_1(p_{i-1}) \quad \implies \quad p_i^{\min} \leq p_{i-1}.$$

Similarly, from the inequality $c_0(\tilde{p}, p_{i-1}) \geq c_i^{\min}$ and the decreasing monotonicity of $s_0(\cdot)$, we can show that $\tilde{p} \leq s_0^{-1}(c_i^{\min} + s_0(p_{i-1})) = p_i^{\max} \geq p_{i-1}$. $\qquad\square$

We shall now provide a detailed, joint proof of Lemmas 3 and 4, for completing the proof of Theorem 1 in Section 3 of the main paper.

**Restatement of Lemma 3.** *If a myopic agent with subjective probability $\pi_i$ and a risk-averse utility function of wealth $u_i(\cdot)$, possessing properties 1, 2, and 3 above, trades with a well-behaved market scoring rule for a single Arrow-Debreu security, and updates the market's instantaneous price from $p_{i-1} \in (0, 1)$ to $p_i$ in the process, then $p_i$ is the unique solution in $(0, 1)$ to the following fixed-point equation:*

$$p_i = \frac{\pi_i u_i'(c_1(p_i, p_{i-1}))}{\pi_i u_i'(c_1(p_i, p_{i-1})) + (1 - \pi_i)u_i'(c_0(p_i, p_{i-1}))}. \tag{4}$$

**Restatement of Lemma 4.** *The implicit function $p_i(p_{i-1}, \pi_i)$ described by (4) has the following properties:*

1. *$p_i = \pi_i$ if and only if $\pi_i = p_{i-1}$.*
2. *$0 < \min\{p_{i-1}, \pi_i\} < p_i < \max\{p_{i-1}, \pi_i\} < 1$ whenever $\pi_i \neq p_{i-1}$, $0 < \pi_i, \neq p_{i-1} < 1$.*
3. *For any given $p_{i-1}$ (resp. $\pi_i$), $p_i$ is a strictly increasing function of $\pi_i$ (resp. $p_{i-1}$).*

*Proof.* If agent $i$'s subjective probability of $\{X = 1\}$ is $\pi_i \in (0, 1)$ and her utility function is $u_i(\cdot)$, her expected myopic utility for taking a trading action that updates the market price $p_{i-1}$ to any $p \in [0, 1]$ is given by

$$\widetilde{u}(p; p_{i-1}, \pi_i) = \pi_i u_i(c_1(p, p_{i-1})) + (1 - \pi_i)u_i(c_0(p, p_{i-1})).$$

The first and second derivatives of the above with respect to $p$ respectively simplify to

$$\widetilde{u}'(p; p_{i-1}, \pi_i) = G''(p) f(p; p_{i-1}, \pi_i);$$
$$\widetilde{u}''(p; p_{i-1}, \pi_i) = G'''(p) f(p; p_{i-1}, \pi_i) + G''(p) f'(p; p_{i-1}, \pi_i),$$

where

$$f(p; p_{i-1}, \pi_i) = \pi_i (1 - p) u_i'(c_1(p, p_{i-1})) - (1 - \pi_i) p u_i'(c_0(p, p_{i-1})) \quad \text{so that}$$
$$f'(p; p_{i-1}, \pi_i) = -\left[ \pi_i u_i'(c_1(p, p_{i-1})) + (1 - \pi_i) u_i'(c_0(p, p_{i-1})) \right]$$
$$+ G''(p) \left[ \pi_i u_i''(c_1(p, p_{i-1}))(1 - p)^2 + (1 - \pi_i) u_i''(c_0(p, p_{i-1})) p^2 \right]$$
$$< 0, \quad \forall p \in (0, 1), \text{ given any } \pi_i, p_{i-1} \in (0, 1),$$

since $G''(\cdot) > 0$, $u_i'(\cdot) > 0$, and $u_i''(\cdot) < 0$ everywhere. Hence, $f(\cdot)$ is strictly decreasing everywhere, its values at $p_{i-1}$ and $\pi_i$ being given by

$$f(p_{i-1}; p_{i-1}, \pi_i) = (\pi_i - p_{i-1}) u_i'(0); \tag{5}$$
$$f(\pi_i; p_{i-1}, \pi_i) = \pi_i (1 - \pi_i) \left[ u_i'(c_1(\pi_i, p_{i-1})) - u_i'(c_0(\pi_i, p_{i-1})) \right]. \tag{6}$$

**Case I** $p_{i-1} < \pi_i$: From (2),

$$c_1(\pi_i, p_{i-1}) - c_0(\pi_i, p_{i-1}) = G'(\pi_i) - G'(p_{i-1}) > 0$$

due to the increasing monotonicity of $G'(\cdot)$. But

$$c_1(\pi_i, p_{i-1}) > c_0(\pi_i, p_{i-1}) \implies u_i'(c_1(\pi_i, p_{i-1})) < u_i'(c_0(\pi_i, p_{i-1}))$$

due to the decreasing monotonicity of $u_i'(\cdot)$. Hence, from (6), $f(\pi_i; p_{i-1}, \pi_i) < 0$.

Also, from (5), since $u_i'(0) > 0$, $f(p_{i-1}; p_{i-1}, \pi_i) > 0$.

These values, along with the decreasing monotonicity of $f(\cdot)$, imply that $f(p; p_{i-1}, \pi_i)$ has a unique zero in $(p_{i-1}, \pi_i)$.

**Case II** $p_{i-1} = \pi_i$: From (5) or (6),

$$f(\pi_i; p_{i-1}, \pi_i) = f(p_{i-1}; p_{i-1}, \pi_i) = 0,$$

and $\pi_i = p_{i-1}$ is the unique zero of $f(p; p_{i-1}, \pi_i)$ due to its monotonic nature.

**Case III** $p_{i-1} > \pi_i$: By symmetry, we can argue exactly as for **Case I** that $f(p; p_{i-1}, \pi_i)$ has a unique zero in $(\pi_i, p_{i-1})$.

Thus for any $\pi_i, p_{i-1}$, there exists a unique solution in $(0, 1)$, say $p^*$, to the equation $f(p; p_{i-1}, \pi_i) = 0$. Since $|G''(p^*)|, |G'''(p^*)| < \infty$, we must have

$$\widetilde{u}'(p^*; p_{i-1}, \pi_i) = 0;$$
$$\widetilde{u}''(p^*; p_{i-1}, \pi_i) = G''(p^*) f'(p^*; p_{i-1}, \pi_i) < 0,$$

since $G''(p^*) > 0$ and $f'(p^*; p_{i-1}, \pi_i) < 0$. In other words, rational risk-averse agent $i$'s price-update $p_i = \arg\max_{p \in [0,1]} \widetilde{u}(p; p_{i-1}, \pi_i)$ is given by $p_i = p^*$ so that

$$f(p_i; p_{i-1}, \pi_i) = 0$$
$$\Rightarrow \quad \pi_i (1 - p_i) u_i'(c_1(p_i, p_{i-1})) = (1 - \pi_i) p_i u_i'(c_0(p_i, p_{i-1})), \quad \text{from definition}$$
$$\Rightarrow \quad \frac{p_i}{1 - p_i} = \frac{\pi_i}{1 - \pi_i} \cdot \frac{u_i'(c_1(p_i, p_{i-1}))}{u_i'(c_0(p_i, p_{i-1}))} \tag{7}$$
$$\Rightarrow \quad p_i = \frac{\pi_i u_i'(c_1(p_i, p_{i-1}))}{\pi_i u_i'(c_1(p_i, p_{i-1})) + (1 - \pi_i) u_i'(c_0(p_i, p_{i-1}))}$$

The last step facilitates the interpretation of $p_i$ as a risk-neutral probability. However, for most subsequent proofs, we shall recall the more convenient odds ratio formulation provided in (7).

Moreover, it is easy to see that the findings in **Case I**, **Case II**, and **Case III** above jointly imply properties 1 and 2 in the theorem statement. To prove property 3, first note that, for $x \in \{0, 1\}$,

$$\frac{\partial}{\partial \pi_i} c_x(p_i(p_{i-1}, \pi_i), p_{i-1}) = s'_x(p_i)\frac{\partial p_i}{\partial \pi_i} = G''(p_i)(x - p_i)\frac{\partial p_i}{\partial \pi_i},$$

$$\frac{\partial}{\partial p_{i-1}} c_x(p_i(p_{i-1}, \pi_i), p_{i-1}) = s'_x(p_i)\frac{\partial p_i}{\partial p_{i-1}} - s'_x(p_{i-1})$$

$$= G''(p_i)(x - p_i)\frac{\partial p_i}{\partial \pi_i} - G''(p_{i-1})(x - p_{i-1})$$

Now, taking the partial derivative with respect to $\pi_i$ of both sides of (7),

$$\frac{1}{(1-p_i)^2}\frac{\partial p_i}{\partial \pi_i} = \frac{1}{(1-\pi_i)^2}\frac{u'_i(c_1)}{u'_i(c_0)} + \left(\frac{\pi_i}{1-\pi_i}\right)\frac{u''_i(c_1)\frac{\partial c_1}{\partial \pi_i}u'_i(c_0) - u'_i(c_1)u''_i(c_0)\frac{\partial c_0}{\partial \pi_i}}{(u'_i(c_0))^2}$$

$$\Rightarrow v_1\frac{\partial p_i}{\partial \pi_i} = v_2 + u''_i(c_1)u'_i(c_0)G''(p_i)(1-p_i)\frac{\partial p_i}{\partial \pi_i} + u'_i(c_1)u''_i(c_0)G''(p_i)p_i\frac{\partial p_i}{\partial \pi_i}$$

$$\text{where } v_1 = \left(\frac{1-\pi_i}{\pi_i}\right)\left(\frac{u'_i(c_0)}{1-p_i}\right)^2, \ v_2 = \frac{u'_i(c_0)u'_i(c_1)}{\pi_i(1-\pi_i)},$$

$$\Rightarrow \frac{\partial p_i}{\partial \pi_i} = \frac{v_2}{v_1 - G''(p_i)\left[u''_i(c_1)u'_i(c_0)(1-p_i) + u'_i(c_1)u''_i(c_0)p_i\right]}$$

$$> 0.$$

This is because $0 < \pi_i, p_i < 1$, $u'_i(c_1), u'_i(c_0), G''(p_i) > 0$, and $u''_i(c_1), u''_i(c_0) < 0$ in our model so that $v_1, v_2 > 0$, hence both the numerator and denominator are positive.

Similarly, taking the partial derivative with respect to $p_{i-1}$ of both sides of (7),

$$\frac{1}{(1-p_i)^2}\frac{\partial p_i}{\partial p_{i-1}} = \left(\frac{\pi_i}{1-\pi_i}\right)\frac{u''_i(c_1)\frac{\partial c_1}{\partial p_{i-1}}u'_i(c_0) - u'_i(c_1)u''_i(c_0)\frac{\partial c_0}{\partial p_{i-1}}}{(u'_i(c_0))^2}$$

$$\Rightarrow v_1\frac{\partial p_i}{\partial p_{i-1}} = u''_i(c_1)u'_i(c_0)\left[G''(p_i)(1-p_i)\frac{\partial p_i}{\partial \pi_i} - G''(p_{i-1})(1-p_{i-1})\right]$$

$$+ u'_i(c_1)u''_i(c_0)\left[G''(p_i)p_i\frac{\partial p_i}{\partial \pi_i} - G''(p_{i-1})p_{i-1}\right]$$

$$\Rightarrow \frac{\partial p_i}{\partial p_{i-1}} = \frac{-G''(p_{i-1})\left[u''_i(c_1)u'_i(c_0)(1-p_{i-1}) + u'_i(c_1)u''_i(c_0)p_{i-1}\right]}{v_1 - G''(p_i)\left[u''_i(c_1)u'_i(c_0)(1-p_i) + u'_i(c_1)u''_i(c_0)p_i\right]}$$

$$> 0$$

for the same reasons as $\frac{\partial p_i}{\partial \pi_i}$.

Hence $p_i(p_{i-1}, \pi_i)$ is increasing in each of $p_{i-1}$ and $\pi_i$, the other remaining constant. $\qquad\square$

**Corollary 1.** *If $\pi_i > p_{i-1}$ (resp. $\pi_i < p_{i-1}$), then $p_{i-1} < p_i < \pi_i$ (resp. $\pi_i < p_i < p_{i-1}$), i.e. a myopic risk-averse agent moves the market price in the direction of her belief but not all the way.*

This intuitive result follows from the analysis in **Case I** and **Case III** of the above proof.

**Corollary 2.** *The agents' beliefs as well as the market's initial price put bounds on the instantaneous price at the end of every episode:*

$$\min\{p_0, \pi_1, \pi_2, \dots, \pi_i\} \le p_i \le \max\{p_0, \pi_1, \pi_2, \dots, \pi_i\}, \quad \forall i = 1, 2, \dots.$$

For the logarithmic market scoring rule (LMSR),

$$c_1(p_i, p_{i-1}) = b\ln\left(\frac{p_i}{p_{i-1}}\right), \qquad c_0(p_i, p_{i-1}) = b\ln\left(\frac{1-p_i}{1-p_{i-1}}\right)$$

so that equation (7) can becomes

$$\frac{p_i}{1-p_i} = \frac{\pi_i}{1-\pi_i} \cdot \frac{u'_i\left(b\ln\left(\frac{p_i}{p_{i-1}}\right)\right)}{u'_i\left(b\ln\left(\frac{1-p_i}{1-p_{i-1}}\right)\right)}. \tag{8}$$

## 2.1 LMSR as LogOP for CARA utility agents

The following is the proof of Theorem 2 from Section 3.1 of the main paper.

**Restatement of Theorem 2.** *If myopic rational agent $i$, having a subjective belief $\pi_i \in (0,1)$ and a risk-averse utility function satisfying criteria 1, 2, and 3 in Section 2 above, trades with a LMSR market with parameter $b$ and current instantaneous price $p_{i-1}$, then the market's updated price $p_i$ is identical to a logarithmic opinion pool between the current price and the agent's subjective belief, i.e.*

$$p_i = \pi_i^{\alpha_i} p_{i-1}^{1-\alpha_i} \Big/ \left[ \pi_i^{\alpha_i} p_{i-1}^{1-\alpha_i} + (1-\pi_i)^{\alpha_i} (1-p_{i-1})^{1-\alpha_i} \right], \quad \alpha_i \in (0,1), \tag{9}$$

*if and only if agent $i$'s utility function is of the form*

$$u_i(c) = \tau_i \left( 1 - \exp\left( -c/\tau_i \right) \right), \quad c \in \mathbb{R} \cup \{-\infty, \infty\}, \text{ for some constant } \tau_i \in (0,\infty), \tag{10}$$

*the aggregation weight being given by $\alpha_i = \frac{\tau_i/b}{1+\tau_i/b}$.*

*Proof.* **Sufficiency:** If agent $i$'s utility is of the form specified in the theorem, then the first and second derivatives of the utility function are respectively

$$u_i'(c) = \exp\left( -c/\tau_i \right) > 0, \quad \text{and}$$
$$u_i''(c) = -\exp\left( -c/\tau_i \right) / \tau_i < 0 \quad \forall c \in [-\infty, \infty].$$

Hence, Lemma 3 is applicable. Making appropriate substitutions in (8),

$$\frac{p_i}{1-p_i} = \frac{\pi_i}{1-\pi_i} \cdot \frac{\exp\left( -\frac{b}{\tau_i} \ln\left( \frac{p_i}{p_{i-1}} \right) \right)}{\exp\left( -\frac{b}{\tau_i} \ln\left( \frac{1-p_i}{1-p_{i-1}} \right) \right)} = \left( \frac{\pi_i}{1-\pi_i} \right) \left( \frac{p_i}{p_{i-1}} \right)^{-b/\tau_i} \left( \frac{1-p_i}{1-p_{i-1}} \right)^{b/\tau_i}$$

$$\text{Thus,} \quad \left( \frac{p_i}{1-p_i} \right)^{1+b/\tau_i} = \left( \frac{\pi_i}{1-\pi_i} \right) \left( \frac{p_{i-1}}{1-p_{i-1}} \right)^{b/\tau_i}$$

Exponentiating both sides by $\frac{1}{1+b/\tau_i}$,

$$\frac{p_i}{1-p_i} = \left( \frac{\pi_i}{1-\pi_i} \right)^{\frac{1}{1+b/\tau_i}} \left( \frac{p_{i-1}}{1-p_{i-1}} \right)^{\frac{b/\tau_i}{1+b/\tau_i}} = \left( \frac{\pi_i}{1-\pi_i} \right)^{\alpha_i} \left( \frac{p_{i-1}}{1-p_{i-1}} \right)^{1-\alpha_i},$$

where $\alpha_i = \frac{1}{1+b/\tau_i} = \frac{\tau_i/b}{1+\tau_i/b}$. Simplifying, we get the required LogOP formulation in the theorem statement; alternatively, by taking the logarithm on both sides, we obtain the equivalent additive log-odds ratio formulation.

**Necessity:** Since we have restricted ourselves to the class of utility functions satisfying criteria 1, 2, and 3, a utility function that results in a logarithmic opinion pool on interacting with LMSR must satisfy Lemma 3 with

$$p_i = \pi_i^{\alpha_i} p_{i-1}^{1-\alpha_i} \Big/ \left[ \pi_i^{\alpha_i} p_{i-1}^{1-\alpha_i} + (1-\pi_i)^{\alpha_i} (1-p_{i-1})^{1-\alpha_i} \right] \quad \text{for some constant } \alpha_i \in (0,1),$$

or, equivalently, with

$$\frac{\pi_i}{1-\pi_i} = \left( \frac{p_i}{1-p_i} \right)^{\frac{1}{\alpha_i}} \left( \frac{1-p_{i-1}}{p_{i-1}} \right)^{\frac{1-\alpha_i}{\alpha_i}}.$$

Making the requisite substitutions in (8) and simplifying, we see that $u_i'(\cdot)$ must satisfy

$$\left( \frac{p_i}{p_{i-1}} \right)^{\frac{1-\alpha_i}{\alpha_i}} u_i'\left( b \ln\left( \frac{p_i}{p_{i-1}} \right) \right) = \left( \frac{1-p_i}{1-p_{i-1}} \right)^{\frac{1-\alpha_i}{\alpha_i}} u_i'\left( b \ln\left( \frac{1-p_i}{1-p_{i-1}} \right) \right)$$
$$\forall p_i, p_{i-1} \in (0,1) \tag{11}$$

since, owing to the fact that each of $\pi_i$ and $p_{i-1}$ is allowed to attain any value in $(0,1)$, $p_i$ defined as the LogOP above can lie anywhere in $(0,1)$ as well.

Since $0 < \frac{p_{i-1}}{\pi_i}, \frac{1-p_{i-1}}{1-\pi_i} < \infty$, we claim that relation (11) is true if and only $u_i'(\cdot)$ satisfies

$$y^{\frac{1-\alpha_i}{\alpha_i}} u_i'(b\ln(y)) = M_i, \quad \forall y \in (0, \infty), \qquad \text{where constant } M_i = u_i'(0). \tag{12}$$

The sufficiency is obvious. To establish the necessity, suppose there exists a risk-averse utility function satisfying (11) but not (12). Then, there must exist $y_1, y_2 \in (0, \infty)$, such that $y_1 > y_2$ without loss of generality, and

$$h(y_1) \neq h(y_2), \quad \text{where} \quad h(y) = y^{\frac{1-\alpha_i}{\alpha_i}} u_i'(b\ln(y)) \quad \forall y \in (0, \infty).$$

But, if $0 < y_2 < 1 < y_1 < \infty$, we can obtain $\tilde{\pi} = y_2(y_1 - 1)/(y_1 - y_2) \in (0, 1)$ and $\tilde{p} = (y_1 - 1)/(y_1 - y_2) \in (0, 1)$ for which (11) is violated, giving us a contradiction. Thus, any $u_i(\cdot)$ satisfying (11) must also obey

$$h(y_1) = h(y_2) \quad \forall y_1, y_2 : 0 < y_2 < 1 < y_1 < \infty.$$

This also means that for any two values $y_1, y_3 \in (1, \infty)$, and any given $y_2 \in (0, 1)$, we must have $h(y_1) = h(y_2)$ as well as $h(y_3) = h(y_2)$, implying that $h(y_1) = h(y_3) \, \forall y_1, y_3 \in (1, \infty)$. By similar reasoning, we can deduce that $h(y_2) = h(y_4) \, \forall y_2, y_4 \in (0, 1)$. Finally, by the continuity of $h(y)$ at $y = 1$, which in turn follows from the continuity of $u_i'(c)$ at $c = 0$ in our model and the obvious continuity of $y^{\frac{1-\alpha_i}{\alpha_i}}$ at $y = 1$, we arrive at (12).

Now, applying the transformation $c = b\ln(y)$, we obtain the first-order ordinary differential equation

$$u_i'(c) = M_i \exp\left(-\frac{1-\alpha_i}{\alpha_i b} c\right), \quad -\infty \leq c \leq \infty$$

where the extreme values of $c$ have been included for continuity. Solving the above, we get

$$u_i(c) = -\frac{M_i \alpha_i b}{1 - \alpha_i} \exp\left(-\frac{1-\alpha_i}{\alpha_i b} c\right) + C_i, \quad C_i \text{ being the constant of integration}$$

$$= -M_i \tau_i \exp(-c/\tau_i) + C_i, \quad \text{where } \tau_i = \frac{\alpha_i b}{1 - \alpha_i} \implies \alpha_i = \frac{\tau_i/b}{1 + \tau_i/b}$$

$$\equiv \tau_i \left(1 - \exp(-c/\tau_i)\right)$$

since a utility function is strategically equivalent to any positive-affine transformation of itself. $\square$

## 2.2 LMSR as LinOP for an atypical utility with decreasing absolute risk aversion

Here, we present the proof of Theorem 3 from Section 3.2 of the main paper.

**Restatement of Theorem 3.** *If myopic rational agent $i$, having a subjective belief $\pi_i \in (0, 1)$ and a risk-averse utility function satisfying criteria 1, 2, and 3 in Section 2 above, trades with a LMSR market with parameter $b$ and current instantaneous price $p_{i-1}$, then the market's updated price $p_i$ is identical to a linear opinion pool between the current price and the agent's subjective belief, i.e.*

$$p_i = \beta_i \pi_i + (1 - \beta_i)p_{i-1}, \quad \text{for some constant } \beta_i \in (0, 1) \tag{13}$$

*if and only if agent $i$'s utility function is of the form*

$$u_i(c) = \ln(\exp((c + B_i)/b) - 1), \quad c \geq -B_i, \tag{14}$$

*where $B_i > 0$ represents agent $i$'s budget, with the aggregation weight being given by $\beta_i = 1 - \exp(-B_i/b)$.*

*Proof.* If agent $i$'s utility is of the form specified in the theorem, then by Lemma 2, we can obtain the lower and upper bounds on the feasible values of $p_i$ as follows:

$$s_1(p_i^{\min}) = c_i^{\min} + s_1(p_{i-1})$$
$$\Rightarrow \quad b\ln(p_i^{\min}) = -B_i + b\ln(p_{i-1})$$
$$= b\ln(p_{i-1}\exp(-B_i/b))$$
$$= b\ln(p_{i-1}(1-\beta_i)), \quad \text{since } \beta_i = 1 - \exp(-B_i/b)$$
$$\Rightarrow \quad p_i^{\min} = p_{i-1}(1-\beta_i), \quad \text{from the monotonicity of } \ln(\cdot); \tag{15}$$

$$s_0(p_i^{\min}) = c_i^{\min} + s_0(p_{i-1})$$
$$\Rightarrow \quad b\ln(1-p_i^{\max}) = -B_i + b\ln(1-p_{i-1}) = b\ln((1-p_{i-1})\exp(-B_i/b))$$
$$\Rightarrow \quad 1 - p_i^{\max} = (1-p_{i-1})\exp(-B_i/b) = (1-p_{i-1})(1-\beta_i)$$
$$\Rightarrow \quad p_i^{\max} = 1 - (1-p_{i-1})(1-\beta_i) = \beta_i + (1-\beta_i)p_{i-1} \tag{16}$$

**Sufficiency:** For $-B_i \leq c < \infty$,

$$u_i'(c) = \frac{\exp((c+B_i)/b)}{b\left(\exp((c+B_i)/b) - 1\right)} > 0, \quad \text{and}$$

$$u_i''(c) = -\frac{\exp((c+B_i)/b)}{b^2\left(\exp((c+B_i)/b) - 1\right)^2} < 0.$$

Hence we can invoke Lemma 3. Now,

$$\exp\left(\frac{c_1(p_i, p_{i-1}) + B_i}{b}\right) = \exp\left(\ln\left(\frac{p_i}{p_{i-1}}\right) + \frac{B_i}{b}\right)$$
$$= \exp\left(\ln\left(\frac{p_i}{p_{i-1}\exp(-B_i/b)}\right)\right)$$
$$= \frac{p_i}{p_{i-1}(1-\beta_i)}$$
$$\Rightarrow \quad \exp\left(\frac{c_1(p_i, p_{i-1}) + B_i}{b}\right) = \frac{p_i}{p_i^{\min}} \quad \text{from (15)}.$$

$$\text{Similarly,} \quad \exp\left(\frac{c_0(p_i, p_{i-1}) + B_i}{b}\right) = \frac{1-p_i}{1-p_i^{\max}} \quad \text{from (16)}.$$

$$\tag{17}$$

$$\text{Hence,} \quad \frac{u_i'(c_1(p_i, p_{i-1}))}{u_i'(c_0(p_i, p_{i-1}))} = \frac{\frac{1}{b} \cdot \frac{p_i/p_i^{\min}}{p_i/p_i^{\min} - 1}}{\frac{1}{b} \cdot \frac{(1-p_i)/(1-p_i^{\max})}{(1-p_i)/(1-p_i^{\max}) - 1}} = \frac{p_i}{1-p_i} \cdot \frac{p_i^{\max} - p_i}{p_i - p_i^{\min}}.$$

It is precisely for obtaining the above ratio that we require the scaling factor of $1/b$, dependent on the market maker parameter, in the exponential in the utility function. Substituting in (8), and noting that $p_i/(1-p_i) \neq 0$ for $0 < p_{i-1} < 1$, we get

$$1 = \frac{\pi_i}{1-\pi_i} \cdot \frac{p_i^{\max} - p_i}{p_i - p_i^{\min}} \quad \Longleftrightarrow \quad p_i = (1-\pi_i)p_i^{\min} + \pi_i p_i^{\max}$$
$$\Longleftrightarrow \quad p_i = \beta_i\pi_i + (1-\beta_i)p_{i-1},$$

on plugging in the expressions for $p_i^{\min}$ and $p_i^{\max}$ from (15) and (16), and simplifying.

**Necessity:** Since we have restricted ourselves to the class of utility functions satisfying criteria 1, 2, and 3, a utility function that results in a linear opinion pool on interacting with LMSR must satisfy Lemma 3 with $p_i = \beta_i\pi_i + (1-\beta_i)p_{i-1}$ for some constant $\beta_i \in (0,1)$. Making the requisite substitutions in (8) and simplifying, we see that $u_i'(\cdot)$ must satisfy

$$\frac{u_i'\left(b\ln\left(\beta_i\left(\frac{\pi_i}{p_{i-1}}\right) + 1 - \beta_i\right)\right)}{\beta_i + (1-\beta_i)\frac{p_{i-1}}{\pi_i}} = \frac{u_i'\left(b\ln\left(\beta_i\left(\frac{1-\pi_i}{1-p_{i-1}}\right) + 1 - \beta_i\right)\right)}{\beta_i + (1-\beta_i)\left(\frac{1-p_{i-1}}{1-\pi_i}\right)}$$
$$\forall p_{i-1}, \pi_i \in (0,1). \tag{18}$$

Since $0 < \frac{p_{i-1}}{\pi_i}, \frac{1-p_{i-1}}{1-\pi_i} < \infty$, we claim that relation (18) is true if and only $u_i'(\cdot)$ satisfies

$$u_i'(b\ln(\beta_i y + 1 - \beta_i)) = K_i\left(\beta_i + \frac{1-\beta_i}{y}\right), \quad \forall y \in (0, \infty), \tag{19}$$

where constant $K_i = u_i'(0)$, and the (negative) lower bound on the domain of $u_i(\cdot)$ is given by $-B_i = b\ln(1 - \beta_i)$ with $u_i'(-B_i) = \infty^1$.

The sufficiency is obvious. To establish the necessity, suppose there exists a risk-averse utility function satisfying (18) but not (19). Then, there must exist $y_1, y_2 \in (0, \infty)$, such that $y_1 > y_2$ without loss of generality, and

$$g(y_1) \neq g(y_2), \quad \text{where} \quad g(y) = \frac{u_i'(b\ln(\beta_i y + 1 - \beta_i))}{\beta_i + \frac{1-\beta_i}{y}} \quad \forall y \in (0, \infty).$$

But, if $0 < y_2 < 1 < y_1 < \infty$, we can obtain $\tilde{\pi} = y_2(y_1 - 1)/(y_1 - y_2) \in (0, 1)$ and $\tilde{p} = (y_1 - 1)/(y_1 - y_2) \in (0, 1)$ for which (18) is violated, giving us a contradiction. Thus, any $u_i(\cdot)$ satisfying (18) must also obey

$$g(y_1) = g(y_2) \quad \forall y_1, y_2 : 0 < y_2 < 1 < y_1 < \infty.$$

This also means that for any two values $y_1, y_3 \in (1, \infty)$, and any given $y_2 \in (0, 1)$, we must have $g(y_1) = g(y_2)$ as well as $g(y_3) = g(y_2)$, implying that $g(y_1) = g(y_3) \; \forall y_1, y_3 \in (1, \infty)$. By similar reasoning, we can deduce that $g(y_2) = g(y_4) \; \forall y_2, y_4 \in (0, 1)$. Finally, by the continuity of $g(y)$ at $y = 1$, which in turn follows from the continuity of $u_i'(c)$ at $c = 0$ in our model and the obvious continuity of $(\beta_i + (1 - \beta_i)/y)$ at $y = 1$, we arrive at (19).

Now, applying the transformation $c = b\ln(\beta_i y + 1 - \beta_i)$, we obtain the first-order ordinary differential equation

$$u_i'(c) = \frac{K_i\beta_i \exp(c/b)}{\exp(c/b) - (1 - \beta_i)}, \qquad b\ln(1 - \beta_i) \leq c \leq \infty$$

where the extreme values of $c$ have been included for continuity. Solving the above, we get

$$
\begin{aligned}
u_i(c) &= K_i\beta_i(b\ln(\exp(c/b) - (1 - \beta_i)) + C_i), \quad C_i \text{ being the constant of integration} \\
&= K_i\beta_i(b\ln(\exp(c/b) - \exp(-B_i/b)) + C_i), \quad \text{since } -B_i = b\ln(1 - \beta_i) \\
&= K_i\beta_i b\ln(\exp((c + B_i)/b) - 1) + K_i\beta_i(C_i - B_i) \\
&\equiv \ln(\exp((c + B_i)/b) - 1)
\end{aligned}
$$

since a utility function is strategically equivalent to any positive-affine transformation of itself. $\quad\square$

## 3 LMSR with logarithmic utility agents

In this section, we shall explore our idea, mentioned in Section 3.2 of the main paper, that agents with logarithmic utility induce an approximate linear opinion pool in a LMSR market under certain conditions.

**Comparison of utility function** (14) **with logarithmic utility:** The two utility functions under consideration are

$$
\begin{aligned}
u_{\mathrm{atyp}}(c; B, b) &= \ln(\exp((c + B)/b) - 1), \quad c \geq B, \\
u_{\log}(c; w) &= \ln(c + w), \quad c \geq w
\end{aligned}
$$

where constants $B, w \in (0, \infty)$ are the respective budgets. First note that both are strictly increasing and strictly concave functions with decreasing absolute risk aversion. Moreover, $u_{\mathrm{atyp}}(c)$ behaves approximately as a logarithmic utility for small values of $(c + B)/b$ and as a linear utility (corresponding to risk-neutrality) for large values thereof.

$$(c + B)/b \ll 1 \implies u_{\mathrm{atyp}}(c; B, b) \approx \ln(1 + (c + B)/b - 1) = \ln(c + B) - \ln b \equiv \ln(c + B);$$

$$(c + B)/b \gg 1 \implies u_{\mathrm{atyp}}(c; B, b) \approx \ln\exp((c + B)/b) = (c + B)/b \equiv c,$$

Figure 1: Comparison of a logarithmic utility function $u_{\log}(c; B) = \ln(c + B)$, $c \geq -B$, where $B = 1$ is the (positive) budget, with various instances of the atypical decreasing absolute risk aversion utility function (14) $u_{\text{atyp}}(c; B, b) = \ln(\exp((c + B)/b) - 1)$ with the same budget $B = 1$ but different scaling factors $b = 0.1, 1, 10$. Note that for $b = B = 1$, the two functions are very close to each other for small (negative and close to $-B$) values of wealth $c$. From the graphs, it appears to be a reasonable conjecture that the two utility functions are most similar, in the sense that the switch in the nature of (14) from approximately logarithmic to approximately linear occurs at a higher value of wealth, for values of $b$ that are comparable to $B$.

using first order approximations, and applying the fact that a utility function is (strategically) equivalent to any positive affine transformation of itself. We provide a visual contrast of the above utility functions in Figure 1.

**Proposition 1.** *For a myopic agent with a subjective probability $\pi_i \in (0, 1)$ and a logarithmic utility function with budget $w_i \in (0, \infty)$, i.e.*

$$u_i(c) = \ln(w_i + c), \, c \geq -w_i,$$

*the updated instantaneous price of a LMSR market with loss parameter $b$ after interaction with the agent can be written as*

$$p_i = \widehat{p}_i + \Delta, \tag{20}$$

*where $\widehat{p}_i$ is a LinOP of $\pi_i$ and $p_{i-1}$ given by*

$$\widehat{p}_i = (1 - \exp(-\widetilde{w}_i))\pi_i + \exp(-\widetilde{w}_i)p_{i-1}, \quad \widetilde{w}_i = w_i/b,$$

*and the error term is*

$$\Delta = \pi_i(1 - p_i) \sum_{j=2}^{\infty} \frac{1}{j} \left( \frac{p_i^{\max} - p_i}{1 - p_i} \right)^j - (1 - \pi_i)p_i \sum_{j=2}^{\infty} \frac{1}{j} \left( \frac{p_i - p_i^{\min}}{p_i} \right)^j,$$

*with $p_i^{\min} = p_{i-1} \exp(-\widetilde{w}_i)$ and $p_i^{\max} = 1 - (1 - p_{i-1}) \exp(-\widetilde{w}_i)$ being the lower and upper bounds on the price $p_i$ imposed by the budget constraint.*

*Proof.* Proceeding exactly as in the proof of Theorem 3 in Section 2.2, we can deduce the bounds $p_i^{\min} = p_{i-1} \exp(-\widetilde{w}_i)$ and $p_i^{\max} = 1 - (1 - p_{i-1}) \exp(-\widetilde{w}_i)$, $\widetilde{w}_i = w_i/b$ on the feasible market price at the end of trading episode $i$, and hence rewrite

$$\widehat{p}_i = (1 - \pi_i)p_i^{\min} + \pi_i p_i^{\max}.$$

For the logarithmic utility, $u_i'(c) = 1/(c + w_i) > 0$ and $u_i''(c) = -1/(c + w_i)^2 < 0$ for $-w_i \leq c < \infty$ so that Lemma 3 can be invoked, and from (8), we can show that

$$(1 - \pi_i)p_i \ln \left( \frac{p_i}{p_i^{\min}} \right) = \pi_i(1 - p_i) \ln \left( \frac{1 - p_i}{1 - p_i^{\max}} \right). \tag{21}$$

Since $0 < \frac{p_i - p_i^{\min}}{p_i}, \frac{p_i^{\max} - p_i}{1 - p_i} < 1$, we can use the well-known Maclaurin series expansion of the logarithmic function

$$\ln(1 + x) = \sum_{j=1}^{\infty} \frac{(-1)^{j+1} x^j}{j}, \quad -1 < x \leq 1$$

to obtain the following:

$$\ln\left(\frac{p_i}{p_i^{min}}\right) = -\ln\left(1 - \frac{p_i - p_i^{min}}{p_i}\right) = \frac{p_i - p_i^{min}}{p_i} + \underline{\delta}_i;$$

$$ln\left(\frac{1 - p_i}{1 - p_i^{max}}\right) = -\ln\left(1 - \frac{p_i^{\max} - p_i}{1 - p_i}\right) = \frac{p_i^{max} - p_i}{1 - p_i} + \overline{\delta}_i,$$

where $\underline{\delta}_i = \sum_{j=2}^{\infty} \frac{1}{j}\left(\frac{p_i - p_i^{min}}{p_i}\right)^j$, and $\overline{\delta}_i = \sum_{j=2}^{\infty} \frac{1}{j}\left(\frac{p_i^{max} - p_i}{1 - p_i}\right)^j$.

Substituting in Equation (21) and simplifying,

$$p_i = \widehat{p}_i + \Delta_i,$$

where $\widehat{p}_i = (1 - \pi_i)p_i^{\min} + \pi_i p_i^{\max}$, and $\Delta_i = \pi_i(1 - p_i)\overline{\delta}_i - (1 - \pi_i)p_i \underline{\delta}_i$. $\qquad\square$

**Approximation of actual $p_i$ by $\widehat{p}_i$:** If, instead of the Maclaurin series in the above proof of Proposition 1, we had used the first-order approximation $-\ln(1 - x) \approx x$, which is reasonable for $|x| \ll 1$,[2] we would have obtained $p_i \approx \widehat{p}_i$. Informally, the smaller the agent's normalized budget $\widetilde{w}_i$, the smaller the range $[p_i^{\min}, p_i^{\max}]$ of feasible values of $p_i$, hence the smaller the fractions $(p_i - p_i^{\min})/p_i$ and $(p_i^{\max} - p_i)/(1 - p_i)$ are, hopefully leading to a better approximation. But this might not even be necessary for achieving a small magnitude of $\Delta_i$ which is the difference of two terms of comparable orders. On eyeballing the expression for $\Delta_i$, it appears to be roughly two orders of magnitude smaller than $\widehat{p}_i$. Since the exact dependence of the approximation error on the value of $\widetilde{w}_i$ is hard to figure out analytically, we adopt a simulation-based approach towards exploring this relationship, described in Section 3.1. But, before that, we perform a quick sanity check on the approximation under consideration. From (21), it is evident that

$$\lim_{\pi_i \searrow 0} p_i = p_i^{\min} = \lim_{\pi_i \searrow 0} \widehat{p}_i; \qquad\qquad \lim_{\pi_i \nearrow 1} p_i = p_i^{\max} = \lim_{\pi_i \nearrow 1} \widehat{p}_i,$$

indicating that the actual and approximate updated market prices coincide for extreme agent beliefs.

## 3.1 Simulations with LMSR and logarithmic utility agents

We ran $5 \times 9$ sets of 1000 simulations each for getting a rough idea about the quality of the approximation $p_i \approx \widehat{p}_i$. For each simulation, we generated a sequence of $n = 100$ agents defined by their time-invariant belief-budget pairs $\{(\pi_i, w_i)\}_{i=1}^n$. Since the parameter of interest is the normalized budget $\widetilde{w}_i$, the exact value of the LMSR loss parameter $b$ is immaterial, and we set it to 1. We sampled the $\widetilde{w}_i$'s uniformly at random from the interval $[0, \widetilde{w}_{\max}]$, $\widetilde{w}_{\max} \in \{0.1, 0.2, 0.25, 0.5, 0.75\}$. The beliefs were random samples from the distribution $\text{BETA}(p_{true}, 1 - p_{true})$, $p_{true} \in \{0.1, 0.2, \ldots, 0.9\}$. Thus our knowledge model was that there was a "true" underlying distibution, $\Pr(X = 1) = p_{true}$, according to which nature would decide the forecast event $X$ in the future, and each agent had some idiosyncratic noisy version $\pi_i$ of this $p_{true}$, the variability of the agents' beliefs being represented by the above BETA distribution with mean $\frac{\alpha}{\alpha + \beta} = p_{true}$ and pseudo-sample size (confidence) parameter $(\alpha + \beta)$ held constant at 1 ($\alpha$ and $\beta$ denote standard parameters of a BETA distribution). Over the $n$ trading episodes, we computed two price trajectories starting at $p_0 = 0.5$ each, one induced by each agent maximizing her myopic expected logarithmic utility[3], and the other by the approximate price update equation that always

Figure 2: Panel (a): The error measure increases with increasing difference between $p_{true}$ and $p_0 = 0.5$ for any fixed $\widetilde{w}_{\max}$, and also with an increase in $\widetilde{w}_{\max}$ for any given $p_{true}$-value; nevertheless, the error appears to be small even for higher values of $\widetilde{w}_{\max}$ (less than 0.03 for $0.1 \leq p_{true} \leq 0.9$, $\widetilde{w}_{\max} \leq 0.75$). Error bars are not shown since standard errors are consistently two orders of magnitude smaller than corresponding sample means. Panels (b) and (c): Price trajectories for two sample simulations with $\widetilde{w}_{\max} = 0.2$ and $\widetilde{w}_{\max} = 2$ respectively are displayed, $p_{true} = 0.7$ for both. On eyeballing, the path of approximate prices (dashed black) seems quite close to the path of true prices (solid green), more so for the lower value of $\widetilde{w}_{\max}$, as expected. Also note the high price volatility in panel (c) corresponding to the higher agent budgets, which is also understandable since the agent is now closer to being risk-neutral. "Lower bound" (dashed blue curve) and "Upper bound" (dashed red curve) for each trading epsiode $i$ correspond to price bounds $p_i^{\min}$ and $p_i^{\max}$ respectively.

rejects the error term in (20). At the end of each simulation, we evaluated the root-mean-squared deviation between these two price trajectories, and averaged these values over all 1000 simulations in the set to obtain the "mean RMSD between true and approximate price processes" which serves as our error measure for the approximation. We report our results in Figure 2. The main takeaway message is that the approximation seems reasonable for a wide range of values of $p_{true}$ and $\widetilde{w}_{\max}$.

We also studied the dependence of the error measure on the parameter $(\alpha + \beta)$ which is inversely related to the variance of the traders' beliefs. We fixed $p_{true} = 0.7$ and varied $(\alpha + \beta)$ over $\{0.1, 0.25, 0.5, 1, 1.5, 2.5, 5, 7.5, 10\}$. The results are reported in Figure 3. For both $\widetilde{w}_{max} = 0.2$ and 0.5, we see that this error measure peaks at 1 and then drops off slowly as $(\alpha + \beta)$ increases further.

Figure 3: Variation of the approximation error measure in our simulations with respect to the pseudo-sample size (confidence) parameter of the distribution of agent beliefs.

## Footnotes

[1]This constraint is necessary since $\lim_{y \to 0^+} K_i \left( \beta_i + \frac{1 - \beta_i}{y} \right) = \infty$; also note that $K_i$ is positive real-valued since $u_i'(c) \in (0, \infty)$ for $c \in (-B_i, \infty)$.

[2]Note that the relative error of the linear approximation of the logarithmic function, i.e. $\left|\frac{x - f(x)}{f(x)}\right|$, where $f(x) = -\ln(1 - x)$, is at most 10% for $x \leq 0.193$.

[3]For agent $i$, we discretized the possible range of $p_i$, i.e. $[p_i^{\min}, p_i^{\max}]$ in steps of $10^{-4}$, computed the vector of expected logarithmic utility values for these discrete $p_i$ values, and chose the $p_i$-value corresponding to the maximum entry in this vector as the updated price.

# References

Tilmann Gneiting and Adrian E Raftery. Strictly proper scoring rules, prediction, and estimation. *Journal of the American Statistical Association*, 102(477):359–378, 2007.