[Reviews · NeurIPS 2015]

Submitted by Assigned_Reviewer_1

Summary of the paper:

The paper aims to address the question of how to combine beliefs to a central judgement in the case that the beliefs held by agents have an information asymmetry in each agents perspective on the future outcomes of particular stochastic events.

In particular the paper particularly focuses on the concept of Hanson 2003 known as the prediction market based on a logarithmic market scoring rule. The market scoring rule framework is sometimes also known as a market studied in the context of cost function based market makers. The motivation provided for the MSR setting involves financial markets in which market makers or dedicated specialists are provided incentives, typically in the form of rebates to take trades on both sides of the book ie. the place limit and market orders in such a way that they create liquidity in the market to facilitate buying and selling of an asset. In this context it is argued that a MSR effectively acts as a cost function-based market maker always willing to take the other side of a trade with any willing buyer or seller, and re-adjusting its quoted price after every transaction.

The key results of this paper is to develop a new unified understanding of the information aggregation characteristics of a market with risk-averse agents mediated by a MSR, with no regard to how the agents' beliefs are formed. Furthermore, the authors are able to demonstrate that the MSR adjusted market is equivalent to an opinion pool.

The main result from a technical perspective are provided in Theorems 2 and 3. Theorem 2 applies to the setting of myopic agents with utility functions of a particular form given by the CARA constant absolute risk aversion form that produce under certain restrictions the equivalence between the markets updated price and the result formed by a logorithmic opinion pool between the current price and the agents subjective beliefs. Note - no budget constraints were utilised - please comment on the implications of this. Theorem 3 applies to the case of decreasing absolute risk aversion for the myopic agents utility.

In both cases Bayesian formulations are provided which produces a nice link with the Machine learning literature.

Quality:

The paper is a high quality paper with an excellent exposition of existing results and newly developed results. It attacks an existing class of problems in an interesting and non-trivial manner, whilst managing to present the results clearly and succinctly.

Clarity:

The paper is well written and clear. The literature review and contextual background to the work is well developed. All results are clearly defined and explained in detail.

Originality:

Distinct from other frameworks and existing results in behavioral finance the results developed in this paper do not presuppose any kind of generative model for agent signals, and also do not involve an equilibrium analysis. Consequently, this means that they may be used as tools to analyze the convergence characteristics of the market price in non-equilibrium situations with potentially fixed-belief or irrational agents

In particular the work takes the approach that they consider the characteristics of the evolution of the price process instead of the properties of prices in equilibrium. In particular, examining the way in which the microstructure noise induces aggregation even when the agents are not considered in a Bayesian manner.

By construction an MSR will elicit the subjective probabilities of a risk-neutral agent in an incentive compatible manner but in this context the author show that generally MSR's will elicit risk-neutral probabilities when they interact with risk-averse agents.

Significance:

The paper studies a well established trading environment microstructure framework in order to establish relationships between traditional belief aggregation methods such as opinion pools and Bayesian inference in two cases that are important from the perspective of the agents utility functional forms being parameterically known, ie. the widely studied CARA class and the decresing absolute risk aversion case.
Summary: The paper is interesting and well written. It covers new approaches and provides insights to the well studied area of agent based price discovery mechanisms in the context of a financial market with myopic agents who hold subjective beliefs on the future state of the market in terms of the risk neutral price. They show that under certain classes of parametric utility function for such agents then the risk neutral price is equivalent to a pooling rule in a Log MSR market.

Submitted by Assigned_Reviewer_2

The paper studies the relationship between market scoring rules and opinion pools in terms of their information aggregation properties. Market Scoring Rules (MSR) are known to impose truthful behavior when agents are myopic and risk-neutral. Authors show in the current paper that if agents are risk-averse however, the probability they end up reporting is equal to a function reflecting their true belief and the most recent market price. As a result, in this case the MSR is equivalent to an to a valid opinion pool (one that satisfies unanimity, boundedness and monotonicity). Authors then illustrate their result by establishing the equivalence between LMSR and logarithmic opinion pool when agents have CARA utilities, as well as, LMSR and linear opinion pools when agents are budget-bounded.

I believe that the interpretation of the market maker's price updating as a Bayesian learning algorithm is an interesting way of looking at the problem, particularly because it allows us to single out the effect of market mechanism on information aggregation without the need to worry about the agents updating their beliefs in a Bayesian manner.

The paper in general is well-organized and easy to follow. I believe the settings authors chose to illustrate their main result on are of sufficient importance (i.e. LMSR with risk averse or budget bounded agents) at least in the prediction market literature. However, while results do shed light on the often-considered-difficult problem of information aggregation in prediction markets, I don't find them technically challenging.
Summary: The paper establishes the equivalence of market price evolution in market scoring rule-based markets and opinion pools. While the problem is not technically very challenging, I believe the intuition the paper provides about the information aggregation properties of prediction markets in general is interesting. If space permits, the paper can be of interest to a subset of the NIPS community.

Submitted by Assigned_Reviewer_3

Summary: The paper studies a market-scoring-rule (MSR) prediction market: agents arrive one at a time with fixed beliefs and risk-averse utilities and each update the current prediction, getting paid according to a proper scoring rule. The paper shows that, for risk-averse, myopic agents who have fixed predictions/beliefs, the market updates satisfy natural axioms for "opinion pools", which are functions for aggregating a group of predictions. The paper considers some special cases. It also makes some related observations and points, e.g. on a Bayesian interpretation of updating.

Good points: I like that the paper explores some implications of the connection. The LMSR as a log opinion pool for CARA agents is mathematically nice. Maybe/hopefully some of the ideas mentioned in this paper can serve as starting points for more research. Drawbacks: I am skeptical whether the connection here is actually deep/interesting/likely to be fruitful. A few thoughts: (a) the three opinion pool axioms are weak/general, so does Theorem 1 tell us much?, (b) is it reasonable to have subjective, "stubborn" agents whose beliefs do not depend on the current state of the prediction market? (maybe...) (c) are these two examples evidence of a broader connection? It makes sense that the math works out nicely for LMSR and CARA, but I'm left wondering if there are nice connections more generally, especially given all of the cited related work.

Other thoughts: - I like the utility function in section 3.2 (LMSR as LinOP) which has logarithmic behavior for small c and linear for large c. (Line #372 typo: should end "when (c+B_i) << b".) But from reading the paper, it's hard to tell if this is artificially constructed or an interesting and natural utility function. Has it been studied elsewhere? - I like the discussion of the Bayesian interpretation and how the aggregation weights depend on the agent's position in line and risk attitude. - I wonder what would happen if traders are also updating their beliefs based on previous trades. - For Lemma 4, property #3, wouldn't we really like to prove that $p_i$ is a strictly increasing function of $\pi_{i'}$ for every $i' \leq i$ ?
Summary: The main premise (connections between prediction markets and opinion pools) is shown to be at least potentially interesting and the paper raises some interesting ideas along the way.

Author Feedback
Author rebuttal: Assigned_Reviewer_2: 
"(a) the three opinion pool axioms are weak/general, so does Theorem 1 tell us much?"
- We kept assumptions to a minimum for Theorem 1 so as to obtain clean and general equivalence results. With this theorem in place, one can now ask a range of more specific questions (such as those in our Sections 3.1 and 3.2) but the solutions are likely to be analytically more involved and harder to interpret. Further exploration of these questions is definitely an interesting future direction.

"(b) is it reasonable to have subjective, "stubborn" agents whose beliefs do not depend on the current state of the prediction market? (maybe...)" & "I wonder what would happen if traders are also updating their beliefs based on previous trades."
- Our results are actually agnostic to how agents formulate their beliefs and react to market history - they should apply to agents who come to their beliefs in any manner, including through Bayesian inference, etc (although we do not consider strategic repeated trading). Making additional assumptions about how agents formulate and / or update their beliefs, considering repeated trading etc. will add further layers to the core results we have established in this paper. Papers related to aggregation in markets with multi-shot agents that update beliefs mainly focus on market price convergence in equilibrium (citations: Ostrovsky 2012, Iyer et al. 2014); our motivation in this paper is to single out the role actively played by the market price setting mechanism itself.

"(c) are these two examples evidence of a broader connection? It makes sense that the math works out nicely for LMSR and CARA, but I'm left wondering if there are nice connections more generally, especially given all of the cited related work."
- In view of the cited work, our market scoring rule-opinion pool connection is the most general one that we know of. While it is true that the exact forms of the two parameterized families of utility functions we considered in our examples make the problem analytically tractable, we believe they have much broader qualitative implications: CARA in Section 3.1 and the novel decreasing ARA utility in 3.2 can be viewed as representatives of risk averse utility families without and with budget constraints respectively, and illustrate markedly different aggregation characteristics of the same market microstructure for different utilities / strategies of participating agents (two ends of a spectrum where the market aggregation moves from "logarithmic" to "linear" pooling).

"it's hard to tell if this is artificially constructed or an interesting and natural utility function. Has it been studied elsewhere?"
-To the best of our knowledge, the decreasing ARA utility function in our Section 3.2 has not been described before; it arises analytically as the solution to our problem of determining the conditions under which LMSR can act as a LinOP in a non-equilibrium setting - note that our relevant result Theorem 3 is of the "if and only if" type; please refer to the full proof of the theorem in Section 2.2 of our supplementary material. We have also described some interesting properties of this utility function in Section 3 of our supplementary material.  

"For Lemma 4, property #3, wouldn't we really like to prove that $p_i$ is a strictly increasing function of $\pi_{i'}$ for every $i' \leq i$ ?"
- The aim of Lemma 4 is to enable us to invoke Lemma 1 for a recursive proof. Combining these two lemmas, it follows readily that $p_i$ is a strictly increasing function of $\pi_{i'}$ for every $i' \leq i$.

Assigned_Reviewer_4: 
"Note - no budget constraints were utilised - please comment on the implications of this."
- Please refer to comment (c) addressed to Assigned_Reviewer_2 above.